# Comparing Regional Distribution Equity among Doctors in China before and after the 2009 Medical Reform Policy: A Data Analysis from 2002 to 2017

**DOI:** 10.3390/ijerph17051520

**Published:** 2020-02-27

**Authors:** Xiaolin Cao, Ge Bai, Chunxiang Cao, Yinan Zhou, Xuechen Xiong, Jiaoling Huang, Li Luo

**Affiliations:** 1School of Public Health, Fudan University, Shanghai 200032, China; 17111020022@fudan.edu.cn (X.C.); baige@fudan.edu.cn (G.B.); 18111020035@fudan.edu.cn (Y.Z.); 18111020029@fudan.edu.cn (X.X.); 2Faculty of Foreign Language, Weifang Medical University, Weifang 261042, Shandong, China; cherry_988@163.com; 3School of Public Health, Shanghai Jiao Tong University School of Medicine, Shanghai 200025, China

**Keywords:** equity, doctor resources, time trends

## Abstract

Background: Although China began implementing medical reforms in 2009 aimed at fair allocation of the regional distribution of doctors, little is known of their impact. This study analyzed the geographic distribution of doctors from 2002 to 2017. Methods: This study calculated the Gini coefficient and Theil index among doctors in the eastern, central, and western regions (Category 1) of China, and in urban and rural areas (Category 2). The statistical significance of fairness changes was analyzed using the Mann–Whitney U test. Results: The annual growth rates of the number of doctors for the periods from 2002 to 2009 and 2010 to 2017 were 2.38% and 4.44%. The Gini coefficients among Category 1 were lower than those in Category 2, and statistically decreased after the medical reforms (P < 0.01) but continued to increase in Category 2 (P = 0.463). In 2017, the Theil decomposition result of Category 1 was 74.33% for the between-group, and in Category 2, it was 95.22% for the within-group. Conclusions: The fairness among the regional distribution of doctors in Category 1 is now at a high level and is better than that before the reforms. While the fairness in Category 2 is worse than that before the reforms, it causes moderate inequality and is continually decreasing. Overall unfairness was found to be derived from the between-group.

## 1. Introduction

Fairness reflects equal social relations and is a common ideal in human society [1,2]. Individuals increasingly pay attention to fairness due to the continuous accumulation of social wealth [3,4]. Here, the fair allocation of health resources means that every member of society has the same rights to health care services [5]. This entails that there are no related differences between geographically defined populations and that all health care resources are reasonably allocated based on individual health care needs [6]. The World Health Organization claims that a commitment to equity is a core element of the primary health care value system and that the right to life-saving and health-promoting interventions must not be denied for any reason [7]. The reasonable input and allocation of health resources within the health system is conducive to improving accessibility to medical and health services, and thus promoting health among all residents.

The unfair allocation of health resources is not a problem that is specific to any given area. Rather, it is a global issue. For instance, fair health resource allocation is considered the most important aspect among the health assessment indicators used in more than 30 European regions [8]. In addition, studies in many developing countries (e.g., India, Bangladesh, Viet Nam, and Thailand) have discovered that health resources are unfairly distributed among people of different ethnicities, income groups, and regions [9,10,11,12,13].

Since China is characterized by a large population and an extensive land area, regional differences are reflected in the per capita gross domestic product [14] and penetrate all social aspects, including the allocation of doctors and health outcome. In China, doctors are free to choose different cities and medical institutions without government restrictions. In 2018, the number of doctors and the population in China was 3.61 million and 1,39,538 million, respectively. Compared to 2017, the annual growth rate of doctors in 2018 was 6.40% higher than the birth rate of 1.94% [15]. The doctors’ salary payment method in China mainly consists of a mixed mode of pay by disease, total insurance payment, pay-per-item, pay-per-capita, and DRGs [16,17]. This payment method is characterized by doctors’ incomes being determined by the number of services they provide. According to the results of a survey on the salaries of 30,000 doctors in China, the average annual salary of doctors in first-tier cities in 2015 were ¥102,000 and ¥63,000 in fourth-tier cities and below, ¥69,000 in first-tier hospitals, and ¥ 83,000 in third-tier hospitals [18]. Therefore, under the market freedom mechanism, the geographical distribution of doctors is different. The most significant regional differences in China exist between its eastern, central, and western regions, as well as between its urban and rural areas. In 2018, the numbers of doctors per 1000 individuals in the eastern, central, and western regions of China were 4.3, 3.8, and 3.5, respectively, whereas the numbers of doctors for 1000 individuals in urban and rural areas were 4.0 and 1.8, respectively [15]. The average life expectancy in Shanghai, which represents the eastern developed regions, is 83.63 years, whereas that of Xizang, which represents the western regions, is only 68.2 years.

In 2017, the 19th National Congress of the Communist Party of China proposed that Chinese society is largely problematized by a “contradiction between the people’s growing needs for a better life and the development of imbalances and inadequacies.” [19]. There is also a contradiction in the health talents field “between the growing health needs of the people and the inadequate imbalance in the allocation of health human resources” [19]. Unfairness has thus created a substantial contradiction in Chinese society.

Health is a basic human right [20]. Under the premise of fully respecting the market’s decisive role in resource allocation, the government has made efforts under the Chinese socialist market economy system to implement and protect the universal right to health. The Central Committee of the Communist Party of China and State Council confronted this issue by promulgating the Opinions on Deepening the Reform of the Medical and Health System in 2009, thereby officially implementing medical reforms. Therefore, one of important task of the 2009 medical reform policy was to improve the construction of a talent team. The main measure is to increase the total number of health personnel (doctors, nurses, and health technicians) and assess their service quality to satisfy the health service needs of people in various regions, as well as optimizing the allocation of health personnel resources in areas with low economic development, such as the western region and villages [21,22].

Developed regions should strengthen underdeveloped areas through long-term cooperation and transformation. This includes support for health technicians, equipment upkeep, and funding provisions. Provincial governments should also improve their regional health planning efforts, including the integration and sharing of medical resources between regions while focusing on weak links, such as rural and community health resource investments [23]. Each region has achieved thorough development while improving the medical system based on actual conditions. These areas have gradually implemented and improved the distribution of health resources across the country. General Secretary Xi Jinping pointed out that medical and health-system reforms have entered a “deep zone.” Therefore, it is necessary to make these reforms a reality. Chinese socialism has also entered a new era.

Human resources are the major building blocks of the health system, which is the “first resource” in health care reform [24]. As the most important and scarce medical resource, doctors shoulder the responsibility of achieving the goal of universal health. They are responsible for treating the sick and saving lives, in addition to working as a backbone for overall national health. In general, resource allocation is more inequitable among doctors than for other health workers [25,26]. The state has invested a large amount of resources into China’s health system to improve overall populational health. However, the 2009 Chinese medical reforms have now been implemented for eight years; it is time to determine what changes have occurred to promote fairness among doctors. This study examined the doctors (licensed doctors and licensed assistant doctors) using data from the China Health Statistics Yearbook to analyze the changes that have been implemented in the medical reform policy for regional distribution equity among doctors in the eastern, central, and western regions of China and in Chinese urban and rural areas between 2002 and 2017. This was done to determine the levels of fairness both before and after the 2009 medical reforms to provide a basis for future policy formulation, as well as to promote the equality of health status among Chinese residents living in different regions.

## 2. Materials and Methods

### 2.1. Data Resources and Regional Divisions

This study derived all data from the China Health Statistics Yearbooks, which were published by National Health Commission of the People’s Republic of China. All published China Statistics Yearbooks (2002–2017) were considered for analysis. The standards and procedures for data collection have been mandatory by national laws and regulations. The yearbook is an important source of data for the Chinese government to monitor and evaluate medical improvement exhibitions and effects; improve national health, health resource allocation, and utilization of medical services; and formulate national policies. The following is the collection method used for the number of doctors and population in this study: (1) The number of doctors is obtained through a census survey. It is registered annually by national medical and health institutions at all levels to fill the number of doctors to log in to the National Health Statistics Network Direct Reporting System, which is part of the National Statistical Survey System for Health Resources and Medical Services. (2) The population is obtained by the state in different years according to a census or sample survey. Among them, the national population comprehensive survey is conducted every 10 years, and the national 1% population sampling survey is conducted every five years. In the remaining years, a sample survey of population changes in the country was conducted according to a sample size of 1‰, and the total population in various regions of the country was predicted. Therefore, all derived data were considered reliable [27,28].

This study was confined to the 31 provinces, autonomies, and municipalities in mainland China (i.e., excluding the Macao and Hong Kong Special Administrative Regions and Taiwan Province). We considered doctors as individuals who had passed a licensing examination and were registered with a county or higher-level health authority as either licensed doctors or licensed assistant doctors. In China, the doctor’s license clearly indicates the category of doctors. The ability level of a licensed assistant doctor is lower than that of a licensed doctor. The licensed assistant doctor must be under the guidance of a licensed doctor to exercise the right of prescription. We chose doctors to analyze health resources equity according to two classifications (i.e., Category 1 which included the eastern, middle, and western regions of China, and Category 2 which included Chinese urban and rural areas). China’s urban areas are more developed than its rural areas; those in eastern locations are often better than those in central or western areas.

On the basis of the official classification used in the urban-rural stratum, doctors were divided into city and county groups from 1985 to 2004 but were classified according to urban and rural groups after 2005. Urban areas include direct-controlled cities and prefectural-controlled city districts, while rural areas include counties and county-level cities. Eastern China contains 11 provinces or direct-controlled cities, including Beijing, Tianjin, Hebei, Liaoning, Shanghai, Jiangsu, Zhejiang, Fujian, Shandong, and Hainan, while the Midlands include the eight provinces of Shanxi, Jilin, Heilongjiang, Anhui, Jiangxi, Henan, Hubei, and Hunan, and the western area covers the 12 provinces, autonomous regions, and direct-controlled cities of Inner Mongolia, Chongqing, Guangxi, Sichuan, Guizhou, Yunnan, Xizang, Shaanxi, Gansu, Qinghai, Ningxia, and Xinjiang. This study used 2009 as a cut-off point (i.e., 2002–2009 as Period 1 and 2010–2017 as Period 2).

### 2.2. Analysis

The distribution of doctors is organized as the ratio of population to doctors in 2 categories (i.e., Category 1 which included the eastern, middle, and western regions, and Category 2 which included Chinese urban and rural areas) for 15 years. There are several ways to measure fairness in health resources, including the concentration, Atkinson, Gini, and Theil indices among others. The Gini and Theil indices are considered the simplest variation indicators as they do not depend on any additional parameters, while only the numbers of doctors and people are needed. The two indices have been widely used to compare the geographical distributions of doctors among regions or over time [26,29,30]. The Gini coefficient can reflect the overall fairness between regions, and the Theil index can decompose equity for both within- and between-groups; it can also be used to determine whether any inequity is attributable to either group [10,31].

The inequality in the distribution of doctors is measured by using the population-to-doctor ratios and Gini index in this study and using the Theil index to analyze the causes of unfairness between regions. The Mann–Whitney U test was conducted using the SPSS (IBM, Armonk, NY, USA) software to analyze whether the Gini coefficients of the two stages for Categories 1 and 2 were statistically significantly different [32]. The more complicated methods involved in the Gini coefficient and Theil index are described below.

### 2.3. Gini Coefficient

Gini coefficient values are set between 0 and 1. Values closer to 0 indicate better equity. The Gini coefficient produces results according to four levels (i.e., less than 0.2 indicates high equality, 0.2–0.3 indicates moderate inequality, 0.3–0.4 indicates low equality, and anything higher than 0.4 is considered extreme inequality) [33,34]. This study used the following formula when calculating Gini coefficient values:$$G=1-\overset{n-1}{\underset{i=0}{\sum }}\left(Y_{i+1}+Y_{i}\right)\left(X_{i+1}-X_{i}\right)$$

n: Total number of districts

$Y_{i}$: Cumulative percentage of doctors in the *ith* district

$X_{i}$: Cumulative percentage of the population in the *ith* district

### 2.4. Theil Index

The Theil Index is a relative indicator that ranges from 0 to 1, there is no universal standard for judging inequality levels. In general, the smaller the T, the greater the equity [8,35]. This study used the following formula to calculate Theil index values:$$T=\overset{n}{\underset{i=1}{\sum }}P_{i}\;log\frac{P_{i}}{Y_{i}}$$

$P_{i}$: Proportion of every district’s population (accounting for the overall Chinese population)

$Y_{i}$: Proportion of doctors working in each district (accounting for the total number of doctors nationwide)

The total Theil index can be divided into two groups (i.e., between-groups and within-groups). The $T_{inter}$ and $T_{intra}$ formulas are written as follows:$$T=T_{intra}+T_{inter}$$
$$T_{intra}=\overset{k}{\underset{g=1}{\sum }}P_{g}T_{g}$$
$$T_{inter}=\overset{k}{\underset{g=1}{\sum }}P_{g}\ln\frac{P_{g}}{Y_{g}}$$

$T_{intra}$: Degree of doctor allocation equity in the given area

$T_{inter}$: Degree of doctor allocation equity between areas

$T_{g}$: T of the two categories (Category 1 and Category 2)

$P_{g}$ and $Y_{g}$ indicate the same as $P_{i}$ and $Y_{i}$.

## 3. Results

### 3.1. Time Trends of Regional Distribution of the Number of Doctors from 2002 to 2017

The number of doctors per 1000 persons in different regions over time can be found in Figure 1. Rate changes are shown in Table 1. The number of doctors per 1000 persons in China have increased over time. The number of doctors has also increased from 1.47 to 2.44 persons per 1000 (65.99%). The average annual growth rate before the 2009 medical reforms was 2.38% but has since increased to 4.44%.

The difference in the number of doctors per 1000 persons was greater in Category 2 than that in Category 1. The eastern region showed the highest number of doctors, while the central and western regions were similar to one another. The number of doctors were approximately twice as great in urban regions than in rural areas; this difference has gradually widened over time.

In Category 1, the average annual growth rate in the eastern region was lower for Period 2 (3.11%) than for Period 1 (3.43%). However, the average annual growth rate was higher for Period 2 (5.14%) than for Period 1 (2.25%) in the central region. The fastest growing area was the western region. There, the average annual growth rate was higher for Period 2 (5.69%) than for Period 1 (1.84%).

In Category 2, the average annual growth rate in urban areas was lower for Period 2 (4.21%) than for Period 1 (4.69%). Rural areas showed the same trend. That is, the average annual growth was lower for Period 2 (3.41%) than for Period 1 (3.71%).

In the figure, Category 1 refers to the eastern, middle, and western regions of China; Category 2 refers to Chinese urban and rural areas. Nation refers to the entire China. Period 1 refers to the years from 2002 to 2009 and Period 2 refers to the years from 2010 to 2017.

### 3.2. Changes of GINI Index According to Each Category

This study judged the fairness of the geographic distribution of the number of doctors in each category using the Gini index (Figure 2). The Gini indices were all below 0.3 for each category, while fairness was good according to the international classification criteria. Over time, the Gini index for Category 1 showed a significant downward trend as a whole, but that of Category 2 was continually increasing.

Category 1 refers to the eastern, middle, and western regions of China and Category 2 refers to Chinese urban and rural areas. Period 1 refers to the years from 2002 to 2009 and Period 2 refers to the years from 2010 to 2017.

### 3.3. Changes in Maldistribution and Regulation Efficacy

The average Gini coefficient, standard deviation, and rate of change for the number of doctors per 1000 persons in the two categories over the two periods are shown in Table 2. The Gini coefficient of doctors per 1000 persons in Category 1 decreased from 0.1278 in Period 1 to 0.0895 in Period 2 (30% reduction). The Gini coefficient for doctors per 1000 persons in Category 2 increased from 0.2112 in Period 1 to 0.2258 in Period 2 (7% increment).

The Gini coefficient for Category 1 decreased from 0.1399 in 2002 to 0.0771 in 2017 (44.89%), but the Gini coefficient for Category 2 increased by 17.65%. The rate of decrease for the annual Gini coefficient of Category 1 for Period 2 (4.10%) was higher than that for Period 1 (2.11%). The conditions in Category 2 were opposite. That is, the annual Gini coefficient for Period 2 (0.93%) was higher than that for Period 1 (0.70%).

We found that the change in Gini coefficient for Category 1 was statistically significant for both periods (P < 0.001). However, the test results for Category 2 were P > 0.05, meaning that the change in Gini coefficient was not significant.

### 3.4. Theil Index Contribution Rate of the Number of Doctors for Each Category

This study revealed the contribution rate of fairness for the within- and between-groups using the Theil index. Fairness was analyzed for each category over a timeframe of 16 years (2002 to 2017). The contribution rates according to the Thiel index for the number of doctors in each category are shown in Table 3 and Table 4. The Theil index trends were the same as those of the Gini index. The within-group contribution rate was higher than that of the between-group for Category 1. However, the between-group contribution rate gradually increased over time. In most years, the eastern region had a higher rate value than that of the central and western areas. Category 2 showed opposite trends from Category 1, in which the contribution rate of the within-group was lower than that of the between-group; the interregional contribution rate was around 90%.

## 4. Discussion

This was a comprehensive nationwide study on the geographic distribution of doctors’ allocation trends in China. The results indicate that the ratio of doctors to total population has increased annually. This was especially evident after the 2009 medical reforms. Therefore, the opportunities for residents to obtain health services are increasing. This is obviously related to the rapid development of China’s social economy and its medical reform policies. China’s per capita GDP rose from ¥7942 in 2000 to ¥59,660 in 2017. But does the increase in the number of doctors per capita in the eastern, central, western, urban, and rural areas mean that equity in each region is better? The answer is negative because these are two different concepts. The average annual growth rate of the number of doctors per 1000 population in the central and western regions exceeds the eastern region, whereas the average growth rate of the number of doctors per 1000 population in rural areas is lower than that in urban areas. As the annual average growth rate changes over time, the results show that the Gini coefficient of the eastern, central, and western regions is shrinking, and the urban and rural Gini coefficients are increasing annually. The change in the growth rate of doctors per capita can be seen from the trend of the broken line in Figure 1.

Some studies in Portugal, Mexico, and Australia have also witnessed this phenomenon, which demonstrated an increasing density of doctors within their populations, and that the overall geographical unfairness has not decreased [31,36,37]. Doctors are likely to concentrate in developed locations and are more inclined to choose coastal areas and big cities. This phenomenon is well explained by the standard position hypothesis theory, which assumes that the free market mechanism does not fail with respect to physician location behavior. According to standard location theory, as the number of physicians increases, the diffusion of physicians from the center to the periphery will spontaneously occur with the decrease in their income. The standard economic theory assumes that doctors seek to maximize profits, and therefore tend to choose higher-income areas. Other scholars assume that doctors’ maximum utility is not only profit, but also non-economic quality-of-life factors (e.g., being near other high-income people, large general hospitals, university-affiliated hospitals, and achieving perfect social public welfare); these factors are concentrated in developed regions and metropolises [32].

The difference in the geographical distribution of doctors will inevitably lead to an imbalance in distribution; therefore, we also used the Gini coefficient to analyze the fairness of the geographical distribution of doctors in each region. Results indicated that fairness in the eastern, central, and western regions was of “high equality,” but that fairness in urban and rural areas was of “moderate equality.” The changes in equity detected in each region are discussed below.

Since 2002, the fairness of the geographical distribution of doctors in the eastern and western regions is increasing every year, especially after the 2009 medical reform. The fairness among the eastern and western regions, determined through the Mann–Whitney U test before and after the reform, is also statistically significant, thus indicating that China is making substantial efforts to narrow the gaps in health care access among people living in these regions. The results show that population growth in the middle and western regions has been steady since 2009, but the number of doctors per 1000 persons has increased rapidly. This is obviously related to the medical reform policy of strengthening underdeveloped areas through long-term cooperation and transformation in the developed areas. This includes support for health technicians, equipment maintenance, and funding provisions. Other studies have also come to similar conclusions [8].

However, is the number of doctors among the eastern, central and western regions fair? The results of the Theil index showed that rate values in the eastern, central, and western regions have been annually decreasing, but the highest rate value was found in the central region in 2002, which was converted to the highest in the eastern region in 2017. This means that internal unfairness has gradually been transferred from the central area to the east. In 2005, the central government proposed the “Rise of Central China” strategy, which is a major task for the state to implement and involves promoting regional coordinated development. In the medical and health field, it is reflected in the efforts to attract health talents to the central regions, including the backward provinces in the central region. This strategy can explain this phenomenon very well. Another finding of the results also indicated that the causes of unfairness in the eastern, central, and western regions mainly came from the within-group rather than the between-group. Related studies have also found that internal differences in the eastern region were important factors that affected internal differences in all three regions [38]. The worst instances of unfairness were found in the eastern part of China’s most developed economy. This issue should be taken seriously. Here, China’s social and economic environments and 2009 medical reform policy can very effectively explain this phenomenon. The level of economic development and amount of health resources in Beijing and Shanghai are much higher than other provinces in the eastern region, while there are also more doctors per capita in Fujian and Hainan than indicated by the region average. Second, the implementation of the medical reform focused more on increasing the number of doctors in the central and western regions, thus ignoring the relatively backward eastern regions. At the same time, the developed provinces and cities (such as Beijing and Shanghai) in the east maintain inherently high-speed development models, thus, leading to increasingly greater differences in those areas.

Another interesting phenomenon is that although the density of doctors annually increased in both urban and rural areas, the Gini coefficient increased rapidly while the fairness level decreased in these areas after the 2009 medical reforms. The Theil index indicated that the contribution rate between these groups was around 90%, thus, indicating that unfairness is mainly due to the between-group. After the implementation of the 2009 medical reform policy, China has made several efforts toward fair distribution of doctors in urban and rural areas. For example, the country stabilizes the ranks of rural doctors through compensation and management systems to avoid brain drain. This includes turning rural doctors into proxy producers of government-provided public health and basic medical services. In this way, rural doctors’ incomes would increase through government compensation, and the compensation fund for infrastructural development of rural health institutions would increase. The management system includes the establishment of a supervision system for compensation, the implementation of compensation distribution, and helping rural doctors set up a medical risk sharing mechanism therefore encouraging young doctors to work in rural areas and increasing the number of rural doctors. For example, in 2010, China implemented a project to train directional medical students for work in rural areas free-of-charge at higher medical universities. Directional medical students refer to medical students who sign a post-graduation employment agreement with the school and local governments before enrolling and promise to serve rural areas for six years after graduation. However, there are still effectiveness problems. Many graduates are unwilling to perform at the grassroots level and tend to leave their posts after expiration [39].

However, there is still an insufficient number of doctors in rural; there were 2.44 doctors for every 1000 persons as of 2017. This is not far from China’s 2020 goal of building a medical and health services system containing 2.5 doctors per 1000 residents. However, while there were 3.97 doctors per 1000 persons in urban areas as of 2017, there were only 1.68 in rural areas. This is far from the target. The main reason why China’s medical reform has had a weak effect after implementing these policies is China’s long-standing urban-rural dual structure is that urban areas have access to high-quality educational resources, medical institutions with advanced facilities, convenient transportation facilities, and shopping centers. Contrarily, rural areas are dominated by the smallholder economy and have supportive infrastructure and other living resources that are significantly inferior to those in large cities. Because they need to consider career development prospects, children’s education, and convenient transportation, the harsh medical and professional-development environments in rural areas have led doctors to choose to work in urban areas. The lack of health talents has become one of the main reasons for the low capacity of rural health services. This has also restricted the promotion of a series of reforms, including those designed to integrate county and township health service systems, implement graded medical treatments, and establish family doctor contracting services.

After the health care reform, the fairness of the number of doctors in the eastern, central, and western regions has improved remarkably; nonetheless, it is still unfair between urban and rural areas. To solve the unfair distribution of doctors in various regions, it is necessary to complete the medical reform of the health sector. The reasons for the remarkable results in the eastern, central, and western regions include not only the effective implementation of the medical reform policy, but also the rapid social and economic development in the central and western regions. The fundamental solution for improving the fairness of doctors in urban and rural areas is to develop the rural economy, facilitate rural urbanization, realize positive interactions between urban and rural areas, and resolve the contradictions presented by China’s dual economic structure. This requires the joint efforts of the entire society and various government departments. In addition, the implementation of the medical reform should also focus on the weak provinces in the eastern regions to achieve overall coordinated development.

This study also had some shortcomings. First, the study did not break down the Theil index of the distribution of urban and rural doctors from the provincial level but calculated the Theil index of urban and rural areas according to the eastern, central, and western regions. However, this had no effect on the findings. Some scholars have analyzed urban and rural data from Chinese provinces [40], revealing that the related trends and decompositions were consistent. Second, this study used the 2009 medical reforms as a time node to compare trends from before and after implementation. However, this did not include an analysis of the factors that led to fairness changes. Therefore, our next study would assess socioeconomic variables to increase the theoretical depth of this analysis.

The important findings in this article have also piqued our interest in the analysis of other health resources and health resources allocation, which could be our next research direction.

## 5. Conclusions

This study’s results show that the density of doctors in China has steadily grown over the years. The allocation of the number of doctors in the eastern, central, and western regions and in urban and rural areas is also relatively fair. Implementation of the 2009 medical reforms has accelerated the fairness of these allocations throughout eastern, central, and western regions. However, the level of fairness between urban and rural areas is now decreasing. Existing unfairness in the eastern, central, and western regions is mainly the result of interregional imbalance. The contribution rate of the Theil index in the eastern region is relatively larger than in other areas, thus, indicating that differences are greatest in this area. Finally, unfairness between urban and rural areas can be traced to the between-groups.

## Figures and Tables

**Figure 1 ijerph-17-01520-f001:**
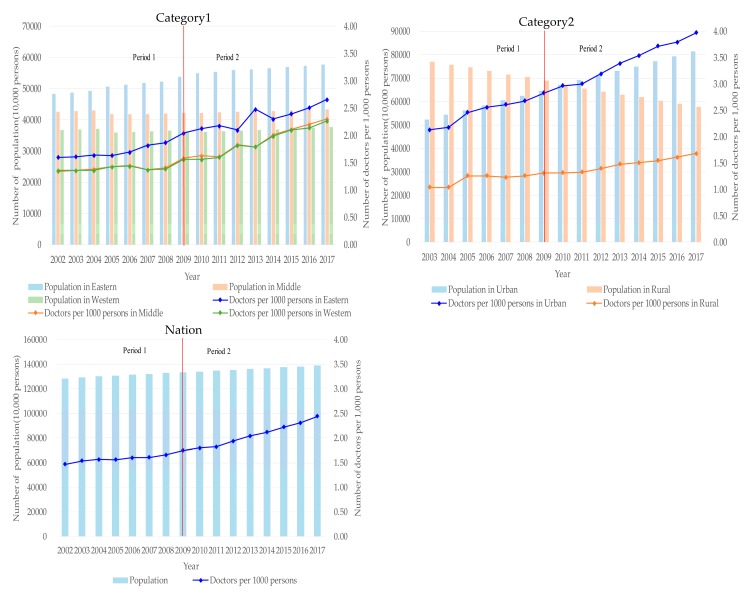
Number of doctors per 1000 persons and populations in different regions in China (2002 to 2017).

**Figure 2 ijerph-17-01520-f002:**
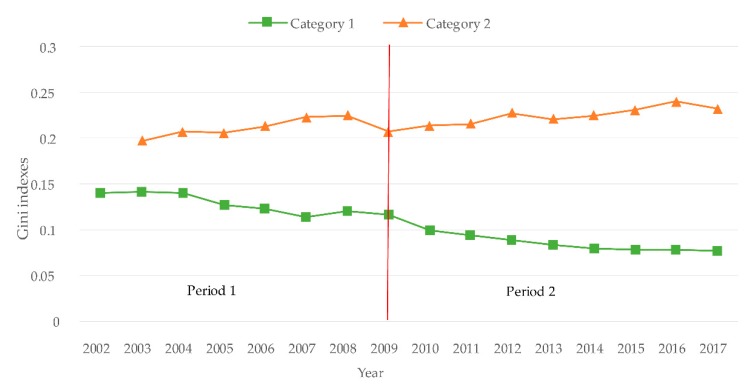
Changes in Gini indices for each category during the period from 2002 to 2017.

**Table 1 ijerph-17-01520-t001:** Changes in number of doctors per 1000 persons in each category.

Variable		Category 1			Category 2		Total
		Eastern	Middle	Western	Urban	Rural	
Number of doctors per 1000 persons	Period 1	1.74	1.42	1.41	2.49	1.20	1.60
	Period 2	2.35	1.94	1.91	3.45	1.49	2.09
Annual change rate (%)	Period 1	3.43	2.25	1.84	4.69	3.71	2.38
	Period 2	3.11	5.14	5.69	4.21	3.41	4.44
Total change rate (%)		66.01	71.16	66.98	86.38	61.54	65.99

Category 1 refers to the eastern, middle, and western regions of China and Category 2 refers to Chinese urban and rural areas. Period 1 refers to the years from 2002 to 2009 and Period 2 refers to the years from 2010 to 2017.

**Table 2 ijerph-17-01520-t002:** Gini scores for each period.

Period		Category 1 Gini Total	Category 2 Gini Total
Total	Year	2002–2017	2003–2017
	N	16	15
	Mean	0.1062	0.2190
	Std. Dev	0.0235	0.0113
	Median	0.1062	0.2213
	Minimum	0.0771	0.1972
	Maximum	0.1417	0.2401
	Change rate (%)	−44.89	17.65
Period 1	Year	2002–2009	2003–2009
	N	8	7
	Mean	0.1278	0.2112
	Std. Dev	0.0107	0.0092
	Median	0.1249	0.2074
	Minimum	0.1135	0.1972
	Maximum	0.1417	0.2251
	Annual change rate (%)	−2.11	0.70
Period 2	Year	2010–2017	2010–2017
	N	8	8
	Mean	0.0895	0.2258
	Std. Dev	0.0123	0.0082
	Median	0.0858	0.2263
	Minimum	0.0782	0.2135
	Maximum	0.1163	0.2401
	Annual change rate (%)	−4.10	0.93
*P*		0.001	0.463

Category 1 refers to the eastern, middle, and western regions of China and Category 2 refers to Chinese urban and rural areas. Period 1 refers to the years from 2002 to 2009 and Period 2 refers to the years from 2010 to 2017.

**Table 3 ijerph-17-01520-t003:** Contribution rates for the Theil index of the number of doctors in Category 1 (2002 to 2017).

Period	Year	Theil L			Decomposition	
		Eastern	Central	Western	Within-group (%)	Between-group (%)
Period 1	2002	0.0133	0.0136	0.0106	89.32	10.68
	2003	0.0132	0.0131	0.0120	89.31	10.69
	2004	0.0118	0.0127	0.0122	87.88	12.12
	2005	0.0121	0.0102	0.0081	91.44	8.56
	2006	0.0108	0.0095	0.0079	87.58	12.42
	2007	0.0115	0.0085	0.0081	86.63	13.37
	2008	0.0109	0.0083	0.0076	85.40	14.60
	2009	0.0083	0.0069	0.0126	90.14	9.86
	Total 1	0.0115	0.0103	0.0099	88.50	11.50
Period 2	2010	0.0063	0.0065	0.0063	90.04	9.96
	2011	0.0056	0.0053	0.0060	86.61	13.39
	2012	0.0046	0.0043	0.0055	81.80	18.20
	2013	0.0042	0.0034	0.0044	77.67	22.33
	2014	0.0040	0.0030	0.0042	78.59	21.41
	2015	0.0044	0.0033	0.0033	77.88	22.12
	2016	0.0046	0.0033	0.0025	76.05	23.95
	2017	0.0046	0.0030	0.0021	74.33	25.67
	Total 2	0.0048	0.0040	0.0043	81.15	18.85
Total		0.0081	0.0072	0.0071	86.22	13.78

**Table 4 ijerph-17-01520-t004:** Contribution rates for Theil index of the number of doctors in Category 2 (2002 to 2017).

Period	Year	Theil L		Decomposition	
		Urban	Rural	Within-group (%)	Between-group (%)
Period 1	2003	0.0003	0.0003	1.06	98.94
	2004	0.0004	0.0001	0.69	99.31
	2005	0.0006	0.0002	1.19	98.81
	2006	0.0008	0.0004	1.53	98.47
	2007	0.0009	0.0004	1.54	98.46
	2008	0.0009	0.0004	1.50	98.50
	2009	0.0007	0.0002	1.24	98.76
	Total 1	0.0007	0.0003	1.27	98.73
Period 2	2010	0.0024	0.0017	5.51	94.49
	2011	0.0027	0.0019	6.24	93.76
	2012	0.0031	0.0023	6.87	93.13
	2013	0.0030	0.0024	7.12	92.88
	2014	0.0024	0.0018	5.15	94.85
	2015	0.0022	0.0014	4.06	95.94
	2016	0.0060	0.0043	11.56	88.44
	2017	0.0015	0.0021	4.78	95.22
	Total 2	0.0029	0.0022	6.45	93.55
Total		0.0019	0.0013	4.07	95.93

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
