# Peer review of "Comparing Regional Distribution Equity among Doctors in China before and after the 2009 Medical Reform Policy: A Data Analysis from 2002 to 2017"

_ijerph, 2020, doi:10.3390/ijerph17051520_

Round 1
Reviewer 1 Report
Thank you for the opportunity to review this paper on the changing distribution of doctors in China. The paper examines the effect of Government reforms implemented in 2009 on doctor population ratios. This was done using the Gini co-efficient and Thiel index. Using a national data source they showed reduced inequality (higher doctor to population ratios) in the Eastern, middle and Western regions of China but growing inequality (lower doctor to population ratios) in urban and rural areas.
The data are based on the China Health Statistics Yearbooks. Presumably these include all doctors in China. We are told that the data used are only for mainland China, presumably for both categories used in the study. A falling doctor to population ratio implies that there has either been a relatively greater increase in population than doctors, or that there has been a relative reduction in the number of doctors in China. If the data are all derived from the national data, which implies that you have the same number of doctors in both categories, how can the doctor population ratio rise in the one category and fall in the other? This seeming anomaly needs to in the paper.
The introduction would be strengthened by providing more information about the health system in China. For example:
Are the doctors allocated to areas?
What method of remuneration is used, state subsidised, insurance, out of pocket, or a mix?
Are the fee tariffs the same in urban and rural areas?
Is the relative production of new doctors exceeding the birth rate?
It would also be helpful to know more about the exactly how the Opinions on Deepening the Reform of the Medical and Health System proposed to reduce inequality. What for example does “"strengthen the construction of a talent team” mean?
Lines 74-75: it is stated that the “resource allocation for doctors is more inequitable among doctors than for other health workers. The reference cited is a 1995 paper. What is the current situation? Is this statement still valid?
Lines 78-82: “This study examined practicing (assistant) physicians using data from the China Health Statistics Yearbook to analyze the current status of and changes that have occurred among resource allocation practices for doctors in the eastern, central, and western regions of China and in Chinese urban and rural areas between 2002 and 2017.” But in lines 92–94, “We considered doctors as individuals who had passed a licensing examination and were registered with a county or higher-level health authority as either licensed doctors or licensed assistant doctors.” So what did the study consider, assistant doctors only or both licensed doctors and licensed assistant doctors?
Also what is meant by, “resource allocation practices for doctors”?
Line 79: What is the difference between physicians and a “practicing (assistant) physicians”? In other parts of the paper physicians are referred to doctors or doctor resources.
Line 118-119: Is this supposed to be, " … to analyze whether the Gini coefficients of the two stages for Categories 1 and 2 were statistically significantly different”.
Lines 154-155: “The total amounts of doctor resources in China have increased over time.” What does this mean - the number of doctors or other resources associated with doctors? The term doctor resources is used in several other places. If it means doctors then this should be corrected throughout.
Table 1: Table 1 and Figure 1 present exactly the same data. The authors should decide which best presents their data: it would appear that the figure is more illuminating.
Figure 2 presents data already presented in Table 2 and as such is redundant.
Line 190: the word decreased in “The Gini coefficient for doctors….decreased from 0.2112 in Period 1 to 0.2258 in Period 2 (7% increment).” should be increased.
Line 197: “We tested whether the two periods were statistically significant over time in each category…”. Surely what was tested was “whether the change in Gini coefficient was statistically significant over time.”?
Line 198: Similarly, “We found that the Gini coefficient for Category 1 was statistically significant for both periods (P < 0.001).” Surely this should be, “…"...the change in Gini co-efficient...".
Line 199-200: “P > 0.05, meaning that the Gini coefficient was not significant.” Presume this should be “the change in Gini coefficient was not significant”?
Figure 3: Again data are repeated in a table and figure. The figure is redundant.
Line 221: “The results indicate that the per-capita consumption of doctor resources has annually increased.” People don't consume doctors. What is probably meant is the ratio of doctors to total population has increased annually.
Lines 227-228: Most similar studies in other countries have demonstrated an increasing density of doctors within their populations. Is this still correct as some countries are now reporting shortages and projected shortages of doctors?
Lines 232-233: “As the number of doctors continues to increase, they will continue to move from the center to the periphery as income levels decrease.” Presumably this refers to the doctors’ income. But the per capita ratio of doctor is falling in urban areas so theoretically there is more work for them.
Line 280: What is a ‘directional’ medical student?
Lines 298-300: “This study was thus only able assess urban and rural areas at the national level. However, the yearbook also divides urban and rural areas among the eastern, central, and western regions.” Are these two sentences not contradictory?
Lines 312-313: “imbalance. “The contribution rate of the Theil index in the eastern region is relatively larger than in other areas, thus indicating that differences are greatest in this area.” Surely The Theil index can't contribute to a difference, it shows the difference?
Reviewer 2 Report
This article is very interesting, of good scientific quality.
3 remarks:
Line 173: A, B, and C represent different regions. Unless I'm mistaken, I can't find the names of these regions. could you include them in figure 1
I understand that it makes more sense to enter Period 1, Period 2 or Category 1, 2. However, this makes it difficult to read the tables and / or figures. could you try to find an alternative for the presentation?
Line 197: We tested whether the two periods were statistically significant over time in each category using a Mann-Whitney U test. To be moved to the Mateials and Methods section
A question: At no time do you speak of demographic change over the period concerned. Why? We do not know the size of the intra-inter-regional population, nor the Physicians / Populations ratio. Could you bring some information or, if it is difficult to tackle this subject in the diuscussion
Reviewer 3 Report
Dear authors,
you study fairness changes in the distribution of doctors in China before the reform in 2009 and after using the Gini and the Theil index.
The topic is really interesting, nevertheless I have some comments.
In your data paragraph exact information abou the sample size is missing. Some descriptives would be important to trust your data. You use in your paper the Gini index and the Theil index. For both general descriptions sources should be mentioned, why they are particular useful for your analysis. Instead of using tests, it could be a better way to calculate confidence intervals (bootstrap/jackknife) to check for significance of your results. At the moment your tests rely on a very small number of observations and what you are actually testing was not always clear to me. This should me specified. As the Goal of your paper is to compare the period before the Reform with period after the Reform I recommend to Focus your tables (Table 1 and Table 5) on the comparison of period 1 to period 2. This would make the tables more informative. Table 1 could than even be combined with Table 2. It was not clear to me why you included Figure 1. This must be better explained. Your paper looks first at differences occured due to the reform and at the interesting aspect of regional differences and differences between urban and rural areas. You should adapt your discussion on these points and really focues on them and explain what we really learn from your analysis. The paragraph about the standard position hypothesis was in this regard not clear to me.Author Response
Please see the attachment.

Reviewer 4 Report
This study judged the fairness of the geographic distribution of doctor resources in China over the 15-year period (2002-2017) encompassing the medical reforms of 2009. Using the Gini and Theil indices, the authors show that, over 2002-2017, the number of doctor per capita has annually increased. Simultaneously, the geographic distribution of medical doctors in the eastern provinces was greater than in the central and western provinces. Urban locations have also been found to be advantaged rural areas in terms of the doctor allocations. Overall, the manuscript is well written and well thought out. The findings are consistent with earlier research documenting changes in health care access across China, such as:
Yip, W., & Hsiao, W. (2009). China's health care reform: A tentative assessment. China economic review, 20(4), 613-619.
Li, X., Cochran, C., Lu, J., Shen, J., Hao, C., Wang, Y., ... & Hao, M. (2015). Understanding the shortage of village doctors in China and solutions under the policy of basic public health service equalization: evidence from Changzhou. The International journal of health planning and management, 30(1), E42-E55.
Fang, P., Dong, S., Xiao, J., Liu, C., Feng, X., & Wang, Y. (2010). Regional inequality in health and its determinants: evidence from China. Health Policy, 94(1), 14-25.
However, I should note that the focus of the study is narrow. The authors examined allocation of doctors and not other health care personnel, and this fact may eschew the entire picture of health resource allocation. For example, rural areas that have fewer doctors may have higher numbers of medical nurses and other medical personal which may compensate the deficit of doctors. I recommend that the authors consider this comment not as a limitation but as a segue to future research.
Round 2
Reviewer 1 Report
The authors have addressed my queries.
One new query, line 131: to what does “The gold content” refer?
The additions to the text need English editing in places.
Author Response
Dear Reviewer 1,
We appreciate the careful review and constructive suggestions to improve our manuscript (ijerph-659888). Following this letter are the reviewer comments with our responses in a different font color. The main changes made to the text are highlighted. All authors approved the final version of the revised manuscript.
Please let us know if there are any other necessary revision. We are looking forward to hear from you soon.
Sincerely,
Xiaoling Cai
Jiaoling Huang
Reviewer 1
Point 1: One new query, line 131: to what does “The gold content” refer?
Response 1: It means the ability level, and I have changed “The gold content” into “The ability level” in line 132.
Point 2: The additions to the text need English editing in places.
Response 2: Thanks for your helpful suggestion. I have revised, and have sought the services of Editage [www.editage.cn] for English language editing. Editage is an international professional language polishing company founded 17 years ago.
Reviewer 3 Report
Dear authors,
the revision improved your paper a lot. However, you should check the language in these new paragraphs again, because there are many grammar mistakes.
The new Paragraph in the introduction (49-67) is from my point of view too Long and parts could be included data section. It might be good to even visualize the Information (descriptives) in a table in the data section.
The result section is much more straightforward than before. Only Table 3 and 4 could be shortenend, as not all Information given in These Tables is needed.
The discussion of your paper improved a lot and gives now clear answers to your Research Questions.
Author Response
Dear Reviewer 3,
We appreciate the careful review and constructive suggestions to improve our manuscript (ijerph-659888). Following this letter are the reviewer comments with our responses in a different font color. The main changes made to the text are highlighted. All authors approved the final version of the revised manuscript.
Please let us know if there are any other necessary revision. We are looking forward to hear from you soon.
Sincerely,
Xiaoling Cai
Jiaoling Huang
Reviewer 3
Point 1: The revision improved your paper a lot. However, you should check the language in these new paragraphs again, because there are many grammar mistakes.
Response 1:Thanks for your helpful suggestion. I have revised, and have sought the services of Editage [www.editage.cn] for English language editing. Editage is an international professional language polishing company founded 17 years ago.
Point 2: The new Paragraph in the introduction (49-67) is from my point of view too Long and parts could be included data section. It might be good to even visualize the Information (descriptives) in a table in the data section.
Response 2: The added content in this paragraph is mainly due to the reviewer 1's proposal, in order to explain the current status of the differences in the income, number and health outcomes of doctors across China. The purpose is to present the theme of our research to demonstrate the severity of regional inequities in China. But the added content is too much, especially the part of the doctors’ income, so we delete the sentence “Generally, the higher the level of medical institutions in developed regions, the larger the number of patients, the more services doctors provide, and the higher the income.” See lines 54-58.
We have shown the number of doctors per 1,000 persons and population in each region in Figure 1, with data for 15 years in a row. So this part is not converted into a table again. The introduction to the doctors’ income is to demonstrate the severity of inequity in China to reveal our research topic, bur not our research results. So it is not added in the data section. In the future, we will study the inequality of income of Chinese doctors in various regions in details.
Point 3:The result section is much more straightforward than before. Only Table 3 and 4 could be shortenend, as not all Information given in These Tables is needed.
Response 3: I have deleted the non-essential data in Table 3 and Table 4.